# Waveguide-integrated twisted bilayer graphene photodetectors

Qinci Wu[1,2,6], Jun Qian[1,2,6], Yuechen Wang[1,2,3,6], Luwen Xing[4,5,6], Ziyi Wei[4], Xin Gao[1,2,3], Yurui Li[2,3], Zhongfan Liu[1,2,3], Hongtao Liu[1], Haowen Shu[4], Jianbo Yin[2,3,4]✉, Xingjun Wang[4]✉ & Hailin Peng[1,2,3]✉

Graphene photodetectors have exhibited high bandwidth and capability of being integrated with silicon photonics (SiPh), holding promise for future optical communication devices. However, they usually suffer from a low photoresponsivity due to weak optical absorption. In this work, we have implemented SiPh-integrated twisted bilayer graphene (tBLG) detectors and reported a responsivity of 0.65 A W$^{-1}$ for telecom wavelength 1,550 nm. The high responsivity enables a 3-dB bandwidth of >65 GHz and a high data stream rate of 50 Gbit s$^{-1}$. Such high responsivity is attributed to the enhanced optical absorption, which is facilitated by van Hove singularities in the band structure of high-mobility tBLG with 4.1° twist angle. The uniform performance of the fabricated photodetector arrays demonstrates a fascinating prospect of large-area tBLG as a material candidate for heterogeneous integration with SiPh.

The capacity of global data communication has been exponentially growing nowadays, calling for new material platforms for data transfer with the capability of high bandwidth, low power consumption, small footprint and low cost[1]. One of the possible solutions in opto-communication is silicon photonics (SiPh), a technology integrating elements such as waveguide, filter, photodetector, and wavelength-division multiplexer into one silicon chip[2–7]. Another advantage of silicon photonics is its ability of hosting other materials (hetero-geneous integration) and enhancing its performance in data transfer[8]. In photodetection, high-mobility materials, such as monolayer graphene, have been integrated with SiPh showing high bandwidth[9–11]. However, one of the main challenges for graphene photodetector is increasing the photoresponsivity. To date, significant efforts have been devoted to enhance the photoresponsivity of graphene SiPh photodetectors, such as integrating graphene with plasmon-assisted nanoarchitectures[9,12], photonic crystal waveguide[13], light-absorbing layer[11], and so on. Until now, these approaches are based on designs of external absorption-enhancing structures instead of a direct design on the core channel material—graphene. Here, we have implemented twisted bilayer graphene (tBLG) into silicon photonics to enhance the optical absorption of the photodetector channel.

The tBLG can be seen as two graphene monolayers that are stacked with a lattice rotation angle $\theta$. It has recently attracted great attention due to its $\theta$-dependent interlayer electronic coupling. At small angles around 1.1°, strong interlayer coupling gives rise to correlated electronic states hosting superconductivity and correlated insulating states[14,15], whereas, at relatively large angles such as 4.1° here, tBLG inherits the linear band structure of high-mobility graphene monolayers but with enhanced optical absorption[16–27] due to the emergence of van Hove singularities (vHs)[28–32]. In our work, this enhancement leads to photo-responsibility up to 0.65 A W$^{-1}$ at a moderate bias of 0.5 V. The high responsivity facilitates the observation of a 3-dB bandwidth of 65 GHz (instrument-limited) and a data stream rate of 50 Gbit s$^{-1}$. In addition, the capability of integrating large-scale high-mobility tBLG into SiPh has been demonstrated by the uniform performance of tBLG photodetectors array, with high responsivities (0.46 ± 0.07 A W$^{-1}$) and high bandwidths (36 ± 2 GHz). This is especially promising given the recent progress

[1]Center for Nanochemistry, Beijing Science and Engineering Center for Nanocarbons, Beijing National Laboratory for Molecular Sciences, College of Chemistry and Molecular Engineering, Peking University, 100871 Beijing, P. R. China. [2]Beijing Graphene Institute, 100095 Beijing, P. R. China. [3]Academy for Advanced Interdisciplinary Studies, Peking University, 100871 Beijing, P. R. China. [4]State Key Laboratory of Advanced Optical Communications System and Networks, School of Electronics, Peking University, 100871 Beijing, P. R. China. [5]School of Engineering, Peking University, 100871 Beijing, P. R. China. [6]These authors contributed equally: Qinci Wu, Jun Qian, Yuechen Wang, Luwen Xing. ✉e-mail: yinjb-cnc@pku.edu.cn; xjwang@pku.edu.cn; hlpeng@pku.edu.cn

in wafer-scale growth of large-size bilayer graphene with controllable twist angle[33] and wafer-scale transfer of graphene to silicon substrate[33,34].

## Results

### Device concept and simulations

The schematic view of the tBLG SiPh photodetector is shown in Fig. 1a, where tBLG is introduced on the Si waveguide as the light-absorbing medium, and a ground-signal-ground (GND-S-GND) electrodes are used to collect photocurrent. This configuration has two electrode/tBLG interfaces at the source terminal (as shown by the two red arrows in the inset of Fig. 1a). Both interfaces have potential gradients and are used for separating photocarriers in tBLG, therefore enhancing the resulting photocurrent[12,35,36]. To target at input light with $\lambda = 1550$ nm, we have calculated the corresponding twist angle $\theta$ as 4.1° by using equation (see Supplementary Fig. 1 for details):

$$\Delta E_{vHs} = 2\hbar\nu_F K \sin\left(\frac{\theta}{2}\right), \qquad (1)$$

where $\Delta E_{vHs}$ is the energy difference between the vHs and Dirac point, $\hbar$ is the reduced Planck's constant, $\nu_F$ is the Fermi velocity for single layer graphene (SLG) and $K = 4\pi/3a_0$ with $a_0$ as lattice constant of SLG[37]. The corresponding band structure and density of states from the continuum model[38] are shown in Fig. 1b, in which red arrows indicate the resonant optical absorption. This resonance accounts for the peak in $\theta$-dependent optical conductivity $\sigma$ relation of tBLG shown as a red line in Fig. 1c, in which the $\sigma$ values for SLG and AB-stacked BLG are also indicated as blue and green lines, respectively (see Supplementary Note 1 and Supplementary Fig. 2).

To evaluate the absorption of tBLG in the device configuration, we have simulated the distribution of incident electric field (see Supplementary Note 2 and Supplementary Fig. 3). In the fundamental quasi-TE mode, the incident electric field intensity at the electrode/tBLG interfaces is particularly intensive which facilitates the absorption of tBLG (see Supplementary Fig. 4). The corresponding absorption of tBLG in this configuration is shown Fig. 1d. With signal-electrode width $W_m$ ~ 200 nm, the absorption coefficients of 4.1° tBLG $\alpha_{tBLG}$ is calculated as 0.58 dB μm$^{-1}$ that is nearly 1.5-times and 3-times higher than AB-stacked BLG (0.39 dB μm$^{-1}$) and SLG ones (0.18 dB μm$^{-1}$). This means that a tBLG device with $L = 10$ μm already absorbs ~60% of incident power. This is higher than AB-stacking BLG (42%), SLG (18%), and those in previous reports[9,12,35,36], suggesting that a much smaller device footprint could be achieved with tBLG as channel material.

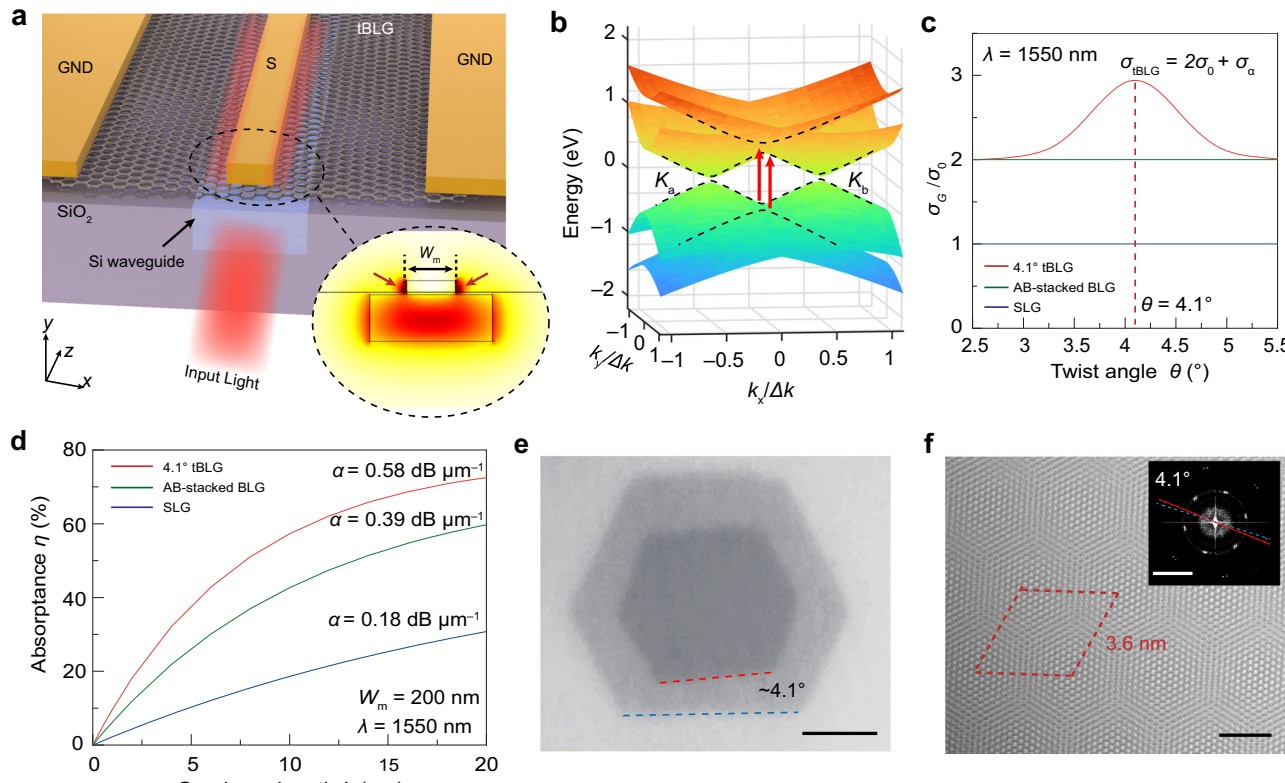

**Fig. 1 | Device concept and simulations of the waveguide-integrated twisted bilayer graphene photodetectors. a** Schematic illustration of a twisted bilayer graphene (tBLG) photodetector. S and GND represent signal and ground electrodes, respectively. Inset: Simulated electric field distribution of the quasi-transverse electric (TE) mode in the waveguide-integrated tBLG photodetector. The red arrows show two electrode/tBLG interfaces at the source terminal, and $W_m$ denotes the width of the signal electrode. **b** Calculated low-energy band structure of 4.1° tBLG. $K_a$ and $K_b$ denote the positions of Dirac points in momentum space. **c** Optical conductivity $\sigma_G$ as a function of twist angle at an incident telecom wavelength of 1550 nm. Different from single-layer graphene (SLG) and AB-stacked bilayer graphene (BLG), the optical conductivity of $\sigma_{tBLG}$ comprises the universal optical conductivity under the linear-band regime $\sigma_0$ and the transition $\sigma_\alpha$ due to the emergence of van Hove singularities. The calculated $\sigma_{tBLG}$ reaches the maximum at 4.1° of twist angle for 1550 nm, shown by the red dashed line. **d** Calculated light absorptance $\eta$ as a function of the propagation length $L$ for 4.1° tBLG, AB-stacking BLG, and SLG, respectively. **e** An optical microscopy (OM) image of a tBLG transferred onto the SiO₂/Si substrate. The twist angle can be roughly determined by the deviation between the straight edges (denoted by dash lines) of two individual graphene monolayers. Scale bar: 5 μm. **f** High-resolution transmission electron microscopic (HR-TEM) image of a tBLG with a clear moiré pattern. The red dash rhomboid indicates the moiré superlattice. Scale bar: 2 nm. Inset: Fast Fourier transform (FFT) of the corresponding HR-TEM image. The twist angle can be precisely determined by the angle between two sets of FFT spots (dash lines). Scale bar: 5 nm⁻¹.

## Structural and electronic properties of tBLG

The tBLG samples were grown by chemical vapor deposition (CVD) on copper foils via a hetero-site strategy[39] and then transferred to target substrates (see the "Methods" section). A sample with domain size of about $10 \times 10\ \mu m^2$ on $SiO_2$/Si substrate is shown in Fig. 1e, in which both component monolayers exhibit hexagonal shapes with sharp edges, suggesting a highly crystalline feature. The interlayer twist angle can be roughly determined as about 4.1° by measuring the angle between two straight edges in the optical image. The evaluation of the crystalline quality and twist angle of tBLG samples can be carried out by high-resolution transmission electron microscopy (HR-TEM, in Fig. 1f), Raman spectroscopy (Supplementary Fig. 5), and photocurrent measurement (Supplementary Note 3 and Supplementary Fig. 6). The HR-TEM image shows a moiré pattern with a period of ~3.6 nm, with a clear signature of lattice orientation that is further confirmed by the Fast Fourier transforms (inset of Fig. 1f). The Raman spectra show a broadening, blueshift, and weak intensity of 2D band, an indicator of small twist angle[33,39]. The Spatially resolved photocurrent mapping of 4.1° tBLG demonstrates an about 4 times larger photocurrent than that of SLG with excitation of 1550 nm light, suggesting its capability in photocurrent enhancement for this wavelength.

High bandwidth of the photodetector requires a low resistance-capacitance (RC) value, which demands high carrier mobility and low contact resistivity. We analyzed these parameters in a Hall device made from ~4.1° tBLG on 300 nm $SiO_2$/Si substrate (Supplementary Note 4 and Supplementary Fig. 7). Room-temperature hole and electron mobilities ($\mu$) are extracted from transport curves as high as 10,600 and 9100 $cm^2\ V^{-1}\ s^{-1}$ respectively, together with a near-zero Dirac voltage $V_D$ of -1.2 V, which corresponds to a small residual carrier concentration of $8.63 \times 10^{10}\ cm^{-2}$. The contact resistivity is estimated as ~500 $\Omega\ \mu m$ (Supplementary Fig. 7), which is comparable to or slightly better than the current level in SLG[40,41]. Given that the device on $SiO_2$/Si is not encapsulated by hBN, the electrical performances of CVD tBLG here are relatively high due to its high crystalline quality and linearly dispersive band structure near Fermi level, suggesting its candidacy for high-performing photodetection material in SiPh.

## Steady-state photoresponse

To demonstrate the photodetection performance of 4.1° tBLG at telecom wavelength of 1550 nm, we transferred the CVD-grown 4.1° tBLG sample (Supplementary Fig. 8) onto an SOI waveguide (thickness about 220 nm, width about 480 nm) and made a photodetector with GND-S-GND configuration (Fig. 2a and Supplementary Fig. 9). Devices with the same geometry but different active channel materials such as AB-stacked BLG and SLG were also fabricated for comparison. As shown in Fig. 2b, tBLG device shows photocurrents $I_{ph}$ of 142 $\mu A$ at 0.5 V bias at incident $\lambda$ of 1550 nm, which is much higher than that of AB-stacked BLG (81 $\mu A$) and SLG (42 $\mu A$), suggesting the advantage of 4.1° tBLG in 1550 nm photodetection. Given the same polarity between the photocurrent and bias, as shown in Supplementary Fig. 10, the dominant photodetection mechanism of the device is ascribed to the photoconductive (PC) effect[42]. This photoresponse enhancement of tBLG is further proven to have relatively wide spectra for telecom photodetection, as shown in Fig. 2c, in which the tBLG photodetector demonstrates a higher $I_{ph}$ than that of AB-stacked BLG and SLG ones ranging from 1510 to 1630 nm. Notably, a photoresponse peak in tBLG device at ~1550 nm originating from the selectively enhanced optical absorption of 4.1° tBLG can be clearly observed, which is distinct from nearly flat photoresponse in AB-stacked BLG and SLG devices.

To evaluate the performance of tBLG device, we measured its photoresponsivity ($R_{ph} = I_{ph}/P_{in}$) at different incident power $P_{in}$ at ($\lambda = 1550$ nm) at 0.5 V bias as shown in Fig. 2d. The $R_{ph}$ reaches 0.65 A W$^{-1}$ at $P_{in} = -9$ dBm, corresponding to an external quantum efficiency (EQE) of 52% and a frequency-normalized noise equivalent

power (NEP) of 46 pW Hz$^{-1/2}$. These values are much better than those of AB-stacked BLG 0.26 A W$^{-1}$ (EQE 21%, NEP 90 pW Hz$^{-1/2}$) and SLG 0.13 A W$^{-1}$ (EQE 10%, NEP 187 pW Hz$^{-1/2}$), respectively (see Supplementary Fig. 9 and Note 6), confirming its advantage in telecom photodetection. These values are much better than those of AB-stacked BLG 0.26 A W$^{-1}$ (EQE 21%, NEP 90 pW Hz$^{-1/2}$) and SLG 0.13 A W$^{-1}$ (EQE 10%, NEP 187 pW Hz$^{-1/2}$), respectively (see Supplementary Figs. 10–12), confirming its advantage in telecom photodetection. We note that the decrease of photoresponsivity with increasing $P_{in}$ is possibly attributed to the nonlinear electron heating process[42,43], which is further discussed in Supplementary Note 6 and Supplementary Fig. 13–14.

To avoid fluctuation of one single device's performance, we fabricated and measured 16 waveguide-integrated graphene photodetectors with the same device geometries (Fig. 2e). The results agree well with our conclusion described above: The tBLG photodetectors yield an average $R_{ph}$ of 0.54 A W$^{-1}$, which is about 3 times and about 6 times higher than those of AB-stacked BLG and SLG photodetectors, respectively. The small $R_{ph}$ deviation of tBLG photodetectors (about 11%) indicates excellent device uniformity by using tBLG as photodetection material. The performance of tBLG devices is also compared with waveguide-integrated graphene photodetectors in recent literatures[9,12,13,35,36,44–55] as shown in Fig. 2f. The responsivity of tBLG photodetectors peers with those of graphene devices with different external enhancement structures. Besides, our photodetectors have a relatively smaller footprint (smaller $L$) due to the enhanced absorption arising from unique vHs of tBLG and the intensive field intensity at the electrode/tBLG interfaces, as shown in Fig. 1d. More impressively, our tBLG photodetectors have a simple device configuration and their performances might be further increased if integrated with additional external structure, such as plasmonic structures[20].

## High-frequency photoresponse and high-speed data transmission

The bandwidths of tBLG photodetectors with small footprints (~8 $\mu m$) were analyzed by using experimental setups as illustrated in Fig. 3a and b (see the "Methods" section). The $S_{21}$ parameter, which represents photoelectrical response, maintains above −3 dB from 1 to 65 GHz (measurement limit of our vector network analysis VNA), indicating that the 3-dB bandwidth ($f_{3dB}$) is beyond 65 GHz (Fig. 3c and Supplementary Fig. 15). Further simulating analysis shows that $f_{3dB}$ in the device might be mainly limited by the RC parameter (see Supplementary Note 8 and Supplementary Fig. 16). Further data transmission measurement shows an eye diagram for 50 Gbit s$^{-1}$ OOK signals, demonstrating the potential of tBLG photodetectors in practical devices (Fig. 3d). The corresponding power consumption (equals to $I_{ph}V_b$/bit rate) of the device is about 0.8 pJ bit$^{-1}$, which is less than typical consumption of a transimpedance amplifier used to amplifier the photocurrent at the backend of a photodetector in SiPh[56,57], suggesting its potential in future applications.

A comprehensive comparison of the tBLG device with state-of-the-art waveguide-integrated graphene photodetectors is shown in Fig. 3e and Supplementary Table 2. The high-frequency response (>65 GHz, limited by instrument) and high responsivity (0.65 A W$^{-1}$) of our tBLG photodetectors outperform most of the other graphene photodetectors. These parameters have already met the requirements of typical commercial photodetectors[58–63], and still have potential for further improvement[11].

## Large-area integration of tBLG photodetectors in SiPh

To demonstrate the possibility of integrating large-scale tBLG into SiPh, we have shown the fabrication and performance of tBLG photodetector array in Fig. 4. The preparation of large-scale tBLG with controllable twist angle is illustrated in Fig. 4a. Unidirectional SLG single crystal on Cu(111)[64] was cut into two pieces and transferred onto

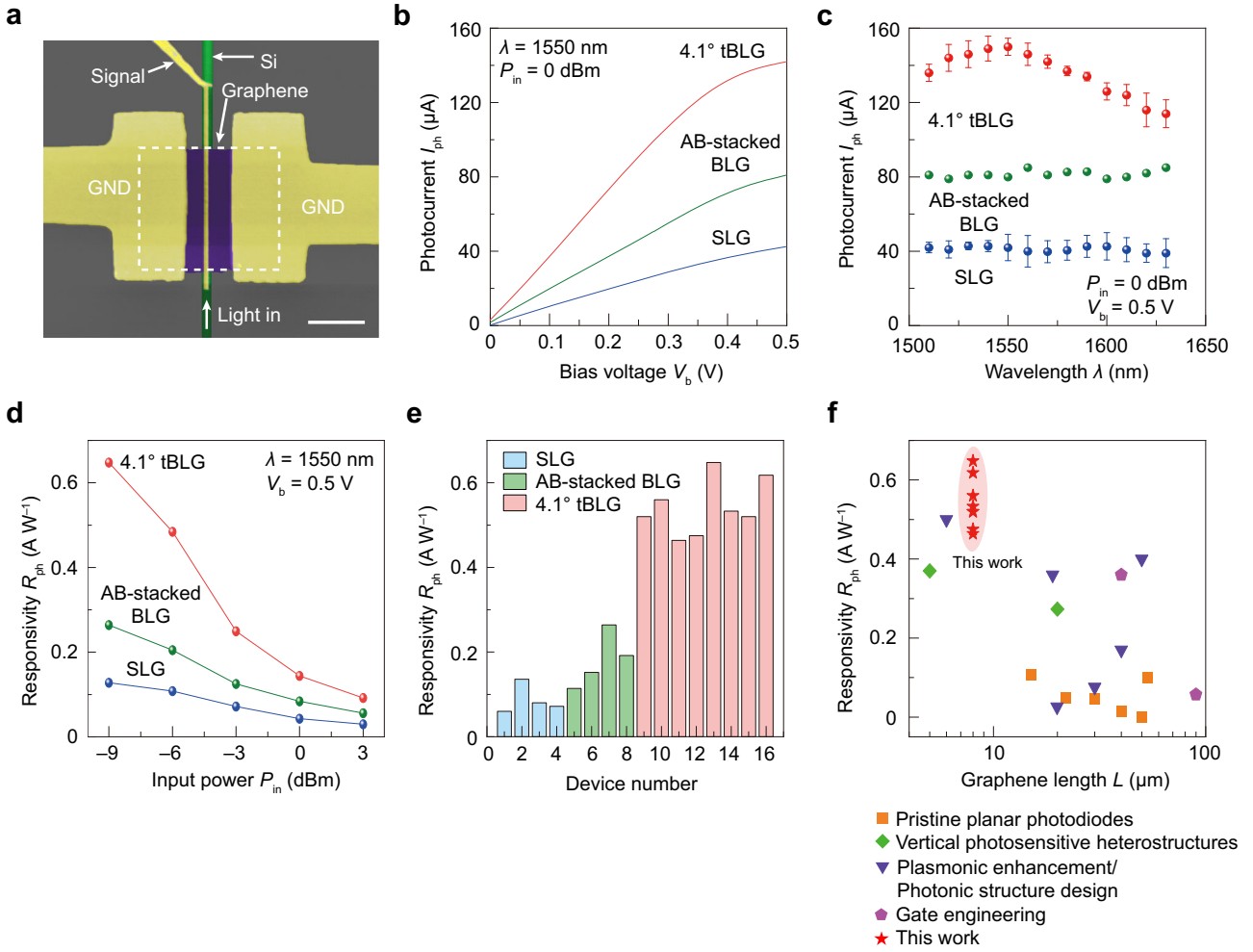

**Fig. 2 | Steady-state photoresponse of photodetectors. a** False color scanning electron microscopic (SEM) image of a waveguide-integrated tBLG photodetector. The white dashed box denotes the region of tBLG. Scale bar: 2 μm. **b** Photocurrent as a function of bias voltage $V_b$ for 4.1° tBLG, AB-stacking BLG, and SLG photodetectors, respectively. **c** Photocurrent as a function of wavelength for 4.1° tBLG, AB-stacking BLG, and SLG photodetectors at a $V_b$ of 0.5 V, respectively. **d** Power dependence of the photoresponsivity at a $V_b$ of 0.5 for 4.1° tBLG, AB-stacking BLG,

and SLG photodetectors, respectively. **e** Statistical histogram of photoresponsivity from 16 devices. Red, green, and blue colors denote 4.1° tBLG, AB-stacking BLG, and SLG photodetectors, respectively. **f** Comparisons of the photoresponsivity versus the graphene length with previously reported waveguide-integrated graphene photodetectors using different optimization strategies, including pristine planar photodiodes[35,36,44,45,50,53], vertical photosensitive heterostructures[48,54], gate engineering[46,47], and plasmonic enhancement[9,12,51]/photonic structure design[13,49,52].

silicon waveguide substrate with a twist angle of about 4.1° (see Fig. 4b and see the "Methods" section). A photodetector array with eight devices was fabricated based on the as-transferred tBLG (Fig. 4c). The devices show an average 3-dB bandwidth of $36 \pm 2$ GHz and an average photoresponsivity of $0.46 \pm 0.07$ A W$^{-1}$, as shown in Fig. 4d and e. In our work, we not only demonstrate a reliable method to fabricate large-scale tBLG and transfer it onto a silicon photonic substrate but also show uniform performances, including responsivity and bandwidth. This confirms the reliability of large-scale integration and fabrication of tBLG.

## Discussion

In summary, tBLG photodetector in SiPh with a small footprint (~8 μm) has shown an outstanding photoresponsivity of 0.65 A W$^{-1}$ for 1550 nm light. This is attributed to the enhanced optical absorption at vHs in the band structure and the intensive incident electrical field at the electrode/tBLG interface, which further facilitates the optical absorption. Benefited from this high sensitivity, the bandwidth extends to >65 GHz (limited by instrument) and 50 Gbit s$^{-1}$ for the OOK data stream. Moreover, a large-area tBLG photodetectors array featuring high

responsivities ($0.46 \pm 0.07$ A W$^{-1}$) and high bandwidths ($36 \pm 2$ GHz) have been demonstrated. Our results show that tBLG with vHs and linearly dispersive band structure can serve as a promising candidate material for a heterogeneous integrated platform for SiPh, especially given the rapid development of wafer-scale growth of tBLG with controllable twist angle and wafer-scale transfer of graphene on silicon substrate[33,34].

## Methods

### Growth and transfer of tBLG

The tBLG samples were prepared on commercially available Cu(111) foils in a low-pressure CVD system. The Cu(111) foil was heated to 1020 °C with 500 sccm Ar and annealed with 500 sccm H$_2$ for 30 min at 1020 °C. Then, the growth of tBLG was carried out under the flow of H$_2$ and CH$_4$ (600:1) at 1020 °C for 30 min. Finally, the system cooled down to room temperature, and the gas flow was turned off.

tBLG was transferred onto the SiPh platform with the assistance of polypropylene carbonate (PPC). The tBLG/Cu was spin-coated with PPC (2000 rpm) and baked at 100 °C for 1 min. After that, PPC/tBLG film was detached from the growth substrate by etching the Cu film for

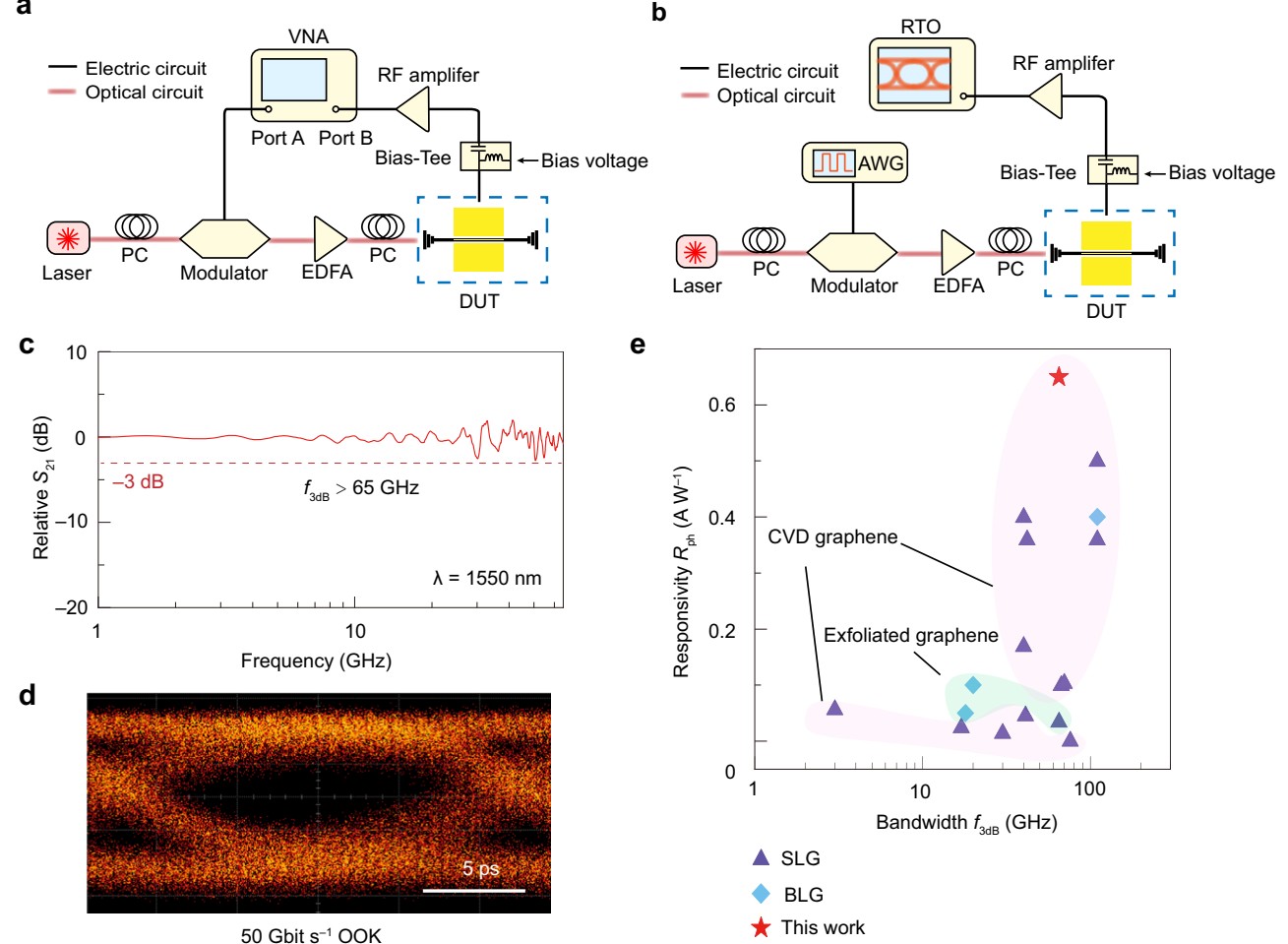

**Fig. 3 | High-frequency photoresponse and data reception testing of waveguide-integrated tBLG photodetectors. a** Experimental set-up for characterizing the 3-dB bandwidth $f_{3dB}$. VNA vector network analyzer, EDFA erbium-doped fiber amplifier, PC polarization controller, RF radio-frequency, DUT device under test. **b** Experimental set-up for detecting the eye diagram. AWG arbitrary waveform generator, RTO real-time oscilloscope. **c** Measured frequency response at a $V_b$ of 0.5 V. 3-dB cutoff is marked by the red dash line. **d** Obtained eye diagram for 50 Gbit s⁻¹ on-off keying (OOK) data stream. **e** Comprehensive comparisons of photoresponsivity and 3-dB bandwidth $f_{3dB}$ of our tBLG device with waveguide-integrated SLG[12,13,36,44–54] and BLG[9,35,44] photodetectors in previous reports.

8–12 h in an aqueous solution of 1 mol L⁻¹ $(NH_4)_2S_2O_8$. A deterministic transfer apparatus and a polydimethylsiloxane (PDMS)/PPC stamp were then used to place the tBLG sample on the waveguide. Finally, the PDMS film was delaminated at 100 °C for 5 min, and the PPC was removed in hot acetone.

Large-area tBLG was prepared as follows: SLG grown on a single-crystal Cu(111) foil was cut into two pieces, one of which was spin-coated with polymethyl methacrylate (PMMA) at 1000 rpm for 1 min and baked at 130 °C for 3 min. After that, the PMMA/graphene/copper foil is etched by 1 mol/L $(NH_4)_2S_2O_8$ solution to remove copper and then rinsed with deionized water. As-prepared PMMA/graphene film is stacked onto the other piece of SLG/Cu(111) foil with a rotation angle. The PMMA/tBLG/copper foil was baked again at 130 °C for 2 h and etched by 1 mol/L $(NH_4)_2S_2O_8$ solution. Finally, the PMMA/tBLG film was transferred onto target substrates.

## Structural characterization of tBLG
The tBLG samples were characterized by optical microscopy (OM, Nikon LV100ND), scanning electron microscopy (SEM, FEI Quattro S field-emission scanning electron microscope operated at 5 kV), transmission electron microscopy (TEM, FEI Tecnai F30 operated at 300 kV and FEI Titan Cubed Themis G2 300 operated at 80 kV), and Raman spectroscopy (633 nm, Horiba HR800).

## Fabrication of Hall-bar devices and four-probe FET measurement
Electron-beam lithography (EBL, FEI Quanta 250 FEG, 30 kV, 350–500 µC cm⁻²) and plasma etching with $O_2$ plasma (Diener Pico) were employed to pattern graphene into a Hall-bar geometry. After that, a methyl methacrylate and methacrylic acid (MMA, MicroChem MMA (8.5) MMA EL6, 4000 rpm)/PMMA (Allresist, 950 K, 4000 rpm) mask was patterned by EBL. Pd/Au (5/50 nm) electrodes were finally deposited by thermal evaporation (ZHD300, Beijing Technol Science Co., Ltd), followed by a standard lift-off technique. Four-probe electrical measurements were carried out with a semiconductor parameter analyzer (Agilent B1500) in a closed-cycle cryogenic probe station (Lakeshore CRX-VF) with a vacuum condition of 10⁻⁵ Torr at room temperature.

## Fabrication of SiPh photodetectors and static photoresponse measurements
The PMMA was spin-coated on the tBLG/SiPh substrate, followed by EBL and $O_2$ plasma etching to pattern tBLG into a channel geometry. After that, the GND-S-GND geometry was exposed to the same EBL process. Pd/Au (5/50 nm) electrodes were deposited by thermal evaporation via an MMA/PMMA mask, followed by a standard lift-off technique.

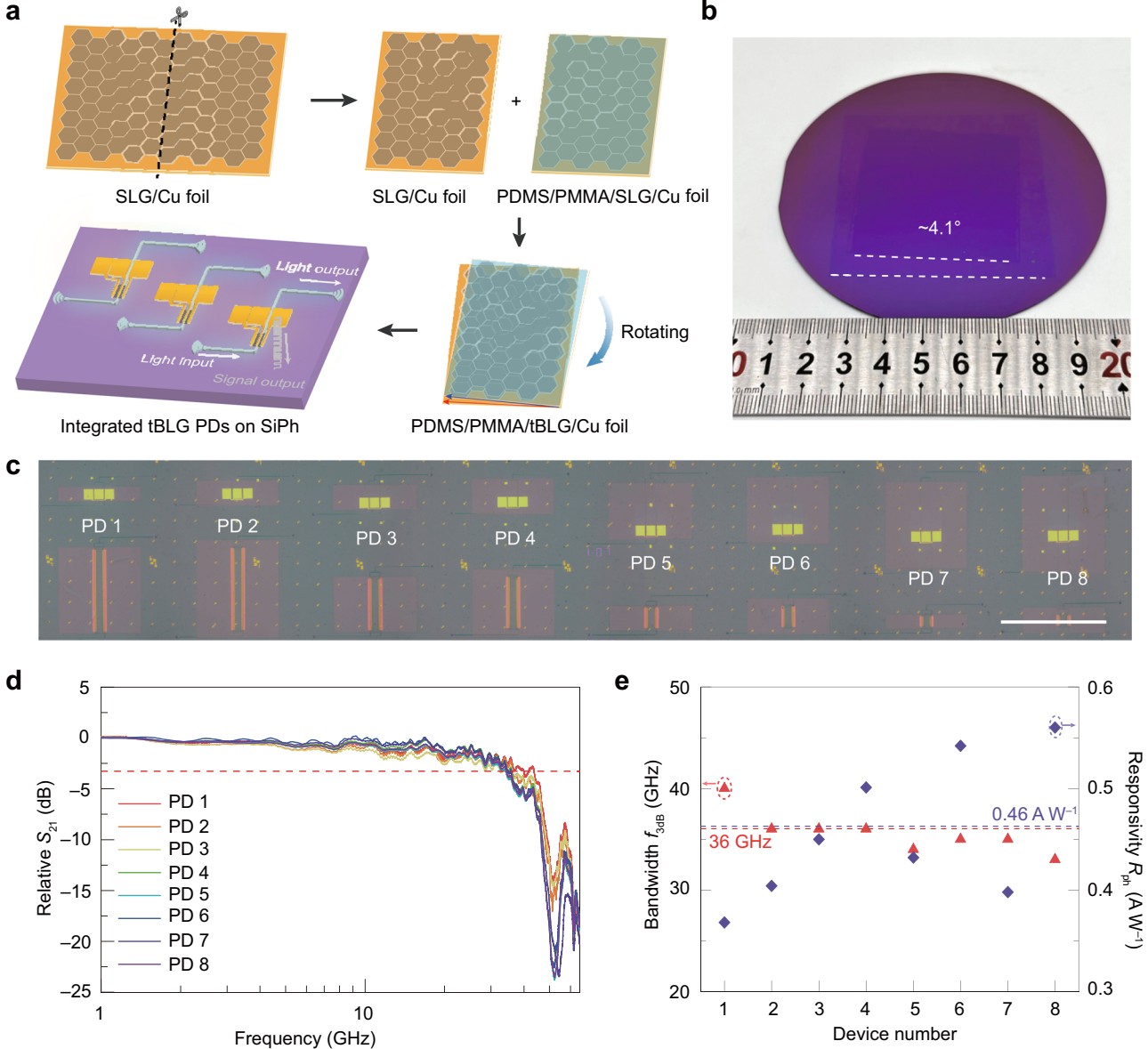

**Fig. 4 | Large-area integration of tBLG photodetectors. a** Schematic illustration of the large-scale transfer of tBLG and fabrication of tBLG photodetectors. PDMS polydimethylsiloxane, PMMA polymethyl methacrylate. **b** Optical image of transferred tBLG on 4-inch SiO$_2$/Si wafer. **c** Optical image of tBLG photodetectors (PDs) array. Scale bar: 1 mm. **d** Measured frequency response of 8 tBLG photodetectors in (**c**). 3-dB cutoff is marked by the red dash line. **e** Statistical histogram of photo-responsivity and 3-dB bandwidth from 8 devices in (**c**). The average bandwidth and responsivity are marked by the red and purple dash lines, respectively.

To perform the static photoresponse characterization of tBLG SiPh photodetectors, a source-meter (Keithley 2400) was used to apply a bias voltage and read the generated photocurrent. A c.w. laser source (Keysight 81606A) was amplified by an erbium-doped fiber amplifier (EDFA) and coupled to the chip with a single-mode optical fiber. A fiber-based polarization controller was used to match the required polarization at the input grating coupler. Before the input grating coupler, a 1:10 optical splitter was used to divide the input light into two light beams in proportion: one output branch of the splitter was coupled to the photodetector, while the other branch was used to monitor the optical power into the grating coupler.

### Broadband measurements

The measurements of the frequency response were performed by a VNA and a LiNO$_3$ Mach–Zehnder modulator (MZM) with a 3-dB bandwidth of 65 GHz. Port A of the VNA was used to drive the MZM, and port B was used to measure the scattering parameter $S_{21}$ of the

tBLG SiPh photodetectors. A bias-tee with 65 GHz bandwidth was used to impose a direct-current (DC) signal. A commercial InGaAs photodetector was used to calibrate the frequency response of the setup.

To measure the optical data transmission, an AWG was used to generate an OOK electrical signal. The electrical signal is modulated onto a 1550 nm optical carrier supplied by a c.w. laser. The optical signal was amplified using an EDFA before launching onto the photodetector. Next, the generated electrical signal was then read out with a high-speed GND-S-GNG probe and recorded by a digital sampling oscilloscope (DSO). The device performance was evaluated by counting the bit errors to determine the BER.

### Data availability

The Source Data underlying Figs. 1d, 2b–d, 3c, 4d and Supplementary Figs. 5a, 6b, c, 7a, b, 11a–c are provided with the paper. All raw data generated during the current study are available from the

corresponding authors upon request. Source data are provided with this paper.

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

## Acknowledgements

This work was supported by the National Key Research and Development Program of China (grant no. 2022YFA1204900), the National Natural Science Foundation of China (grant nos. T2188101, 52021006, 62235002, 62322501), Beijing National Laboratory for Molecular Sciences (BNLMS-CXTD-202001), and the New Cornerstone Science Foundation through the XPLORER PRIZE. We acknowledge the Molecular Materials and Nanofabrication Laboratory (MMNL) in the College of Chemistry at Peking University for the use of instruments.

## Author contributions

H.P., J.Y., Q.W. and J.Q. conceived the original idea for the project and designed and organized the experiments. Y.W. and Z.L. were responsible for the sample preparation of tBLG. Y.W. and X.G. carried out the transfer and structural characterization of tBLG. Q.W. performed the fabrication and four-probe measurement of tBLG Hall bar devices and data analysis. Q.W. and J.Q. carried out the device fabrication of waveguide-integrated tBLG (AB-stacked BLG and SLG) photodetectors. The steady-state and broadband measurements of waveguide-integrated photodetectors were accomplished by Z.W., L.X, H.S. and Y.L. The manuscript was written by H.P., Q.W., J.Q., H.L. and J.Y. with input from other authors. The device simulation and optoelectronic characterization were assisted by J.Y. and X.W. The whole work was supervised by H.P. All authors contributed to the scientific planning and discussions.

## Competing interests

The authors declare no competing interests.

## Additional information

**Peer review information** : *Nature Communications* thanks Hao Wang and the other, anonymous, reviewers for their contribution to the peer review of this work. A peer review file is available.

