## [Peer Review File · Nature Communications]

Waveguide-integrated twisted bilayer graphene
photodetectorsREVIEWER COMMENTS

Reviewer #1 (Remarks to the Author):

The manuscript entitled "Waveguide-integrated twisted bilayer graphene photodetectors" by H Peng et al describes a Twisted bilayer graphene photodetector on Si Photonics waveguide and the authors show that a suitable twisting angle (4.1°) is beneficial for the performance of the detector compared to both single layer graphene photodetector and (AB type) bilayer graphene photodetector. The responsivity is quite high for graphene photodetectors, 0.65A/W @ 1550nm , and the bandwidth exceeds 65GHz even though limited by the instrumentation.

The work is well written, the supplementary information complete the support for understanding the work. The references are complete and give a good overview of the field.

The work is very good and deserves publication in Nature Communications because it is definitely an important advance in the field and the result is very useful for the community.

I have observed that the responsivity is growing by decreasing the input power. It seems that, by further decreasing the input power, the responsivity could be even better. I wonder why the authors didn't try to find the upper limit for the high value of responsivity. Possibly this information or an improved measurement should be included in the revised version.

The second comment is related to the size of the crystal, $8\mu\text{m}$, used for testing and for the devices. What limited the crystal size? The authors should comment on the small size of the graphene crystal they used for the experiments.

Reviewer #2 (Remarks to the Author):

The manuscript under review presents a study on a waveguide-integrated photodetector utilizing twisted bilayer graphene (tBLG). The fundamental concepts regarding the enhanced absorption of twisted bilayer graphene have been extensively investigated and documented in prior research (refs. 16-27). While the device concept of a waveguide-coupled tBLG-based photodetector has been previously proposed and studied (Photonics 2022, 9, 867), this work is reporting an experimental demonstration. The claimed highest reported photo-responsivity of 0.65 A/W may be overstated, as an earlier study reported a slightly higher value of 0.7 A/W (ACS Photonics 2020, 7, 932–940). Additionally, other research has achieved very close and even unsaturated responsivity values, as illustrated in Fig. 2f. The differences in responsivity performance among these devices could be insignificant. Furthermore, the realized speed performance of the device ($\sim 65\text{ GHz}$) may be considered uncompetitive compared to many other graphene waveguide detectors with demonstrated bandwidths of at least up to 110 GHz , as depicted in Fig. 3e or Supplementary Table 1.

Here are some specific comments for further consideration:

1. Elaboration and discussion on the contributing photodetection mechanisms are crucial for a comprehensive understanding of graphene photodetectors in the context of the presented device.

2. Including a figure illustrating the current-voltage characteristics with and without light incidence would provide important insights into aspects such as contact behavior, dark current levels, and would aid in justifying noise figure calculations.
3. It would be beneficial to assess the device performance under negative bias voltages to understand its behavior comprehensively.
4. The observation that the twisted bilayer graphene device exhibits a stronger power dependence on responsivity compared to single and bilayer graphene devices, as shown in Fig. 2d, warrants further discussion.
5. How is the bias dependent speed performance of the device?
6. Clarification is needed regarding the consistently lower bandwidth (~36 GHz) observed in eight large-area integrated tBLG photodetectors (Fig. 4d) compared to the device exhibiting a better bandwidth (~65 GHz) shown in Fig. 3c. Is the device achieving 65 GHz exceptionally fast compared to others?

Reviewer #3 (Remarks to the Author):

Comments to the Author

Qinci has conducted a detailed study on waveguide-integrated twisted bilayer graphene photodetectors, achieving a record-high photoresponsivity of 0.65 A/W for the telecom wavelength of 1550 nm. This breakthrough was made possible by the enhanced optical absorption in twisted bilayer graphene, facilitated by van Hove singularities in the band structure with a 4.1° twist angle. The study showcases significant advancements in photodetector performance, including over 65 GHz bandwidth and 50 Gbit/s data transmission rates, promising for future opto-communications and silicon photonics integration. I find this study to be a particularly intriguing contribution. I recommend this paper for publication, noting its potential impact. However, there are some areas that require further scrutiny and detailed analysis, as highlighted below:

1. The author posits that This configuration has two electrode/tBLG interfaces at source terminal (as shown by the two red arrows in the inset of Fig. 1a). Both interfaces have potential gradients and are used for separating photocarriers in tBLG, therefore double the total photocurrent. This assertion presumes a direct correlation between photocurrent enhancement and graphene's optical absorption capabilities. However, the claim that this configuration doubles the total photocurrent necessitates a more comprehensive discussion. Specifically, it would be beneficial to elucidate the mechanisms through which these interfaces enhance photocarrier separation and photocurrent generation. Additionally, for comparative analysis, could you specify the reference structure or conditions under which this doubling of photocurrent is observed? This clarification would provide a clearer understanding of the efficiency and effectiveness of the proposed design in photodetector applications.
2. Could the author provide a more detailed analysis comparing the absorptive properties of the 4.1° twisted bilayer graphene (tBLG), the AB-stacked bilayer graphene, and single-layer graphene (SLG)? The

manuscript notes that the absorption curve for the 4.1° tBLG reaches saturation more rapidly than those of the other two materials. This observation suggests underlying differences in optical properties and electron-band interactions among these graphene configurations. It would be insightful to understand the mechanisms driving this distinct behavior of the 4.1° tBLG, particularly in terms of its band structure and interaction with light. Further elucidation on how these differences impact the overall performance and efficiency of photodetectors employing these materials would also be valuable.

3. Supplementary Figure 3 is missing essential elements, specifically scale bar/color bar. These components are crucial for interpreting the figure's data accurately. Could the author please address this oversight by adding the missing scale and color bars to Supplementary Figure 3? Including these elements would significantly enhance the figure's clarity and enable readers to better understand the presented data.

4. In the supplementary section, it is stated that 'The contact resistance is estimated as $\sim 500 \Omega \cdot \mu\text{m}$ (Supplementary Fig. 6),' which suggests a unit of resistivity rather than resistance. Typically, contact resistance is measured in ohms (Ω), while ohm-micrometers ($\Omega \cdot \mu\text{m}$) are a unit of resistivity. This discrepancy might lead to confusion about the nature of the measurement being reported. Could the author please clarify this point and review the units used to ensure they accurately represent the quantities being measured? Correcting this could enhance the precision and reliability of the data presented.

5. The statement regarding the photodetectors having a relatively smaller footprint due to enhanced absorption, attributed to the unique van Hove singularities (vHs) of twisted bilayer graphene (tBLG), alongside the intensified field intensity at the electrode/tBLG interfaces as illustrated in Figure 1d, raises a question about the mechanism of field enhancement. Is the field enhancement primarily attributed to the plasmonic mode induced by the interaction between the metal and the waveguide or specifically by the tBLG itself? Clarifying the primary source of field enhancement would provide deeper insight into the operational dynamics of the photodetectors and the role of tBLG in enhancing their efficiency.

Response Letter

Response to the 1st Reviewer

The manuscript entitled "Waveguide-integrated twisted bilayer graphene photodetectors" by H Peng et al describes a Twisted bilayer graphene photodetector on Si Photonics waveguide and the authors show that a suitable twisting angle (4.1°) is beneficial for the performance of the detector compared to both single layer graphene photodetector and (AB type) bilayer graphene photodetector. The responsivity is quite high for graphene photodetectors, $0.65A/W$ @ $1550nm$, and the bandwidth exceeds $65GHz$ even though limited by the instrumentation.

The work is well written, the supplementary information complete the support for understanding the work. The references are complete and give a good overview of the field.

The work is very good and deserves publication in Nature Communications because it is definitely an important advance in the field and the result is very useful for the community.

Response:

We appreciate the reviewer's recognition of the importance of our work. We have addressed the reviewer's comments point by point as follows.

Question 1:

I have observed that the responsivity is growing by decreasing the input power. It seems that, by further decreasing the input power, the responsivity could be even better. I wonder why the authors didn't try to find the upper limit for the high value of responsivity. Possibly this information or an improved measurement should be included in the revised version.

Response:

We thank the reviewer's suggestion. Following the suggestion, we have fabricated a new device and measured the power-dependence of the responsivity, as shown in Fig. R1. Similar with the results in the main text, the responsivity mainly shows decreasing dependence with the input power increasing. This is ascribed to the nonlinear temperature-dependence of electron temperature, which has been observed and studied before [1-5]. We have also found that at lower input power (<-9 dBm, or <0.125 mW), the responsivity seems independent of input power. Although this phenomenon has been observed before [5], its origin hasn't been well studied. We have included Fig. R1 as **Supplementary Fig. 14** in the Supplementary Information to show more information to the community.

Fig. R1 | Power-dependence of the photoresponsivity of 4.1° tBLG photodetector #2. The source-drain bias (V_b) is at 0.5 V.

References:

- [1] M. Freitag et al. *Nat. Photon.* **2013**, 7, 53-59;
- [2] M. W. Graham et al. *Nat. Phys.* **2013**, 9, 103-108;
- [3] P. Ma et al. *ACS Photon.* **2019**, 6, 154-161;
- [4] J. Guo et al. *Light. Sci. Appl.* **2020**, 9, 29;
- [5] S. Schuler et al. *Nat. Commun.* **2021**, 12, 3733)

Question 2:

The second comment is related to the size of the crystal, 8um, used for testing and for the devices. What limited the crystal size? The authors should comment on the small size of the graphene crystal they used for the experiments.

Response:

We thank the reviewer’s comment. Our simulations (Fig. 1d in the main text) indicate that the light absorptance of the twisted bilayer graphene (tBLG) device approaches nearly 60% when the size is $\sim 8 \mu m$. This is reasonable as the absorptance is higher than monolayer and AB-stacked bilayer graphene [1,2,3]. It is also why we chose this domain size: to show crystal orientations with clear twist angle so that the concept can be well demonstrated. The enhanced absorption enables tBLG devices to have smaller footprints which is beneficial for further integration with other circuits. In addition, the tBLG crystal size can be further increased—even to wafer scale—by either on-site

growth method [4] or twist-angle-controlled transfer method. In our paper, latter method has been demonstrated with device arrays and they all show high and uniform responsivity (Fig. 4a in the main text).

References:

[1] P. Ma, et al. *ACS Photon.* **2019**, 6, 154–161.

[2] A. Pospischil, et al. *Nat. Photon.* **2013**, 7, 892–896.

[3] Y. Gao, et al. *Opt. Lett.* **2018**, 43, 1399–1402.

[4] C. Liu, et al. *Nat Mater* **2022**, 21, 1263-1268.

Response to the 2nd Reviewer

The manuscript under review presents a study on a waveguide-integrated photodetector utilizing twisted bilayer graphene (tBLG). The fundamental concepts regarding the enhanced absorption of twisted bilayer graphene have been extensively investigated and documented in prior research (refs. 16-27). While the device concept of a waveguide-coupled tBLG-based photodetector has been previously proposed and studied (Photonics 2022, 9, 867), this work is reporting an experimental demonstration.

Response:

We appreciate the insightful comments from the reviewer. Indeed, tBLG has attracted much attention with intriguing properties, such as the enhanced absorption, and yet the waveguide-coupled tBLG detector only stalls in theoretical concept. In this work, we overcome challenges including building a clean conceptual device, precisely fabrication and large-scale integration silicon photonics. We hope our experience and finding might decrease the barrier between theoretical conceptual and experimental realization, and trigger further investigations. We address the detailed comments as follows.

The claimed highest reported photo-responsivity of 0.65 A/W may be overstated, as an earlier study reported a slightly higher value of 0.7 A/W (ACS Photonics 2020, 7, 932–940). Additionally, other research has achieved very close and even unsaturated responsivity values, as illustrated in Fig. 2f. The differences in responsivity performance among these devices could be insignificant. Furthermore, the realized speed performance of the device (~65 GHz) may be considered uncompetitive compared to many other graphene waveguide detectors with demonstrated bandwidths of at least up to 110 GHz, as depicted in Fig. 3e or Supplementary Table 1.

Response:

We thank the reviewer for the comments on the most important parameters of photodetectors in silicon photonics. Based on the reviewer's comments, we have deleted the "record high" in the sentence of "record high responsivity of 0.65 A W⁻¹ for telecom wavelength 1,550 nm". We would like to point out that the 0.7 A/W (data in the paper actually shows the highest value as 0.67 A/W) reported in *ACS Photonics* **2020**, 7, 932–940 is under mechanism of bolometric effect, which origins from lattice heating and should be much slower than electron heating in our device' mechanism. Therefore, for data communication application which needs fast photoresponse, it will be a real challenge to use the bolometric effect.

Actually, we agree with the reviewer about the insignificance of device performances to some extent, as different labs have different fabricating processes, which leads to different performances. This is also the reason that we focus on a new conceptual material with potential of wafer-scale integration to silicon photonics. The idea is by using a simple device design, the performance has already been outstanding as shown in Figs. 2f and 3e. In addition, we would also like to note that the bandwidth of 65 GHz is the limit of our experimental setup. Higher bandwidth might be achieved based on

P
A
G
E

the theoretical analysis in the Supplementary Note 8.

To make it clear to the readers, the changes we have made are listed as follows:

“...In this work, we have implemented twisted bilayer graphene (tBLG) in SiPh detectors and reported a responsivity of 0.65 A W^{-1} for telecom wavelength 1,550 nm...”
(page 2)

“...In our work, this enhancement leads to photo-responsibility up to 0.65 A W^{-1} at a moderate bias of 0.5 V...” (page 3)

“...In summary, tBLG photodetector in SiPh with small footprint ($\sim 8 \mu\text{m}$) has shown an outstanding photoresponsivity of 0.65 A W^{-1} for 1550 nm light.....” (page 7)

Question 1&2:

Elaboration and discussion on the contributing photodetection mechanisms are crucial for a comprehensive understanding of graphene photodetectors in the context of the presented device.

Including a figure illustrating the current-voltage characteristics with and without light incidence would provide important insights into aspects such as contact behavior, dark current levels, and would aid in justifying noise figure calculations.

Response:

We thank the reviewer’s suggestions. Following the suggestions, we have included the current-voltage curves of different samples with and without light illumination in the as **Supplementary Fig. 10** as shown below. The elaboration and discussion on the mechanism have also been added.

The mechanisms of graphene photodetection include photo-thermoelectric (PTE), photoconductive (PC) and photo-bolometric (PB) effects. PTE effect origins from diffusivity (Seebeck coefficient) differences of hot electrons at different doping levels [1]. It dominates when the source-drain bias is zero or small. In PTE effect, the photocurrent polarity (flowing direction) shouldn’t change with source-drain bias polarity. PC effect origins from the migration of hot electrons under source-drain bias, which dominates at relatively large source-drain bias [2, 3]. In this regime, the polarity of photocurrent should be the same with that of the source-drain bias. As we have explained in the last response, the PB effect origins from resistivity differences at different lattice temperature [2, 3]. In this regime, light illumination heats the lattice, increase the resistivity, and decrease the total current, so the “photocurrent”, defined as difference of current with and without illumination, should show opposite polarity with the source-drain bias. Therefore, with the polarity of photocurrent as key evidence [2] and the same polarity of photocurrent with source-drain bias in **Supplementary Fig. 10**, We conclude that the PC effect is the dominating mechanism in our devices.

We have added the figures and discussion in the manuscript and Supplementary information, the details are shown as follows:

Main manuscript:

Page 5 line 140, we added “...Given the same polarity between the photocurrent and bias as shown in the Supplementary Fig. 10, the dominant photodetection mechanism of the device is the photoconductive (PC) effect....”

Supplementary Information:

Page S10, Supplementary Note 5 is added:

The mechanisms of graphene photodetection include photo-thermoelectric (PTE), photoconductive (PC) and photo-bolometric (PB) effects. PTE effect origins from diffusivity (Seebeck coefficient) differences of hot electrons at different doping levels. It dominates when the source-drain bias is zero or small. In PTE effect, the photocurrent polarity (flowing direction) shouldn't change with source-drain bias polarity. PC effect origins from the migration of hot electrons under source-drain bias, which dominates at relatively large source-drain bias. In this regime, the polarity of photocurrent should be the same with that of the source-drain bias. As we have explained in the last response, the PB effect origins from resistivity differences at different lattice temperature. In this regime, light illumination heats the lattice, increase the resistivity, and decrease the total current, so the “photocurrent”, defined as difference of current with and without illumination, should show opposite polarity with the source-drain bias. Therefore, with the polarity of photocurrent as key evidence and the same polarity of photocurrent with source-drain bias in Supplementary Fig. 10, We conclude that the PC effect is the dominating mechanism in our devices.

Page S22, Supplementary Fig. 10 is added:

Supplementary Fig. 10 | Current-voltage characteristics of graphene PDs. a–c, I – V_b curves under dark and incident power of 3 dBm of 4.1° tBLG (a), SLG (b), and AB-stacked BLG (c) PDs, respectively. Inset: The corresponding I_{ph} of 4.1° tBLG (a), SLG (b), and AB-stacked BLG (c) PDs, respectively.

References:

- [1] N. M. Gabor, et al. *Science* **2011**, 334, 648-652.
- [2] M. Freitag, et al. *Nat. Photon.* **2013**, 7, 53–59.
- [3] F. H. L. Koppens, et al. *Nat. Nanotechnol.* **2014**, 9, 780-793.

Question 3:

It would be beneficial to assess the device performance under negative bias voltages to understand its behavior comprehensively.

Response 3:

We thank the reviewer for the suggestion. We have added the device performance under negative bias voltage as **Supplementary Fig. 11** and the corresponding discussion in **Supplementary Note 5**. The result here echoes with PC mechanism as explained in our response to ‘Question 1&2’. The device performance is almost symmetric under positive and negative bias voltages because the bias voltages act as photocurrent extracting “forces” and should extract similar scale but opposite polarities. The other notable feature is the nonlinear dependence of photocurrent (responsivity) with bias voltage. This is due to that with voltage increasing the extracting efficiency increases and the main bottle neck lies on the photocurrent numbers which is mainly dependent on incident power [1].

Thanks to the suggestion, we have added the figure as **Supplementary Fig. 11** and the discussion in to **Supplementary note 5**.

Supplementary Fig. 11 | Steady-state characterization of graphene PDs. a–c, $R_{ph}-V_b$ curves at different incident power P_{in} of 4.1° tBLG (a), SLG (b), and AB-stacked BLG (c) PDs, respectively.

References:

- [1] M. Freitag, et al. *Nat. Photon.* **2013**, 7, 53–59.

Question 4:

The observation that the twisted bilayer graphene device exhibits a stronger power dependence on responsivity compared to single and bilayer graphene devices, as shown in Fig. 2d, warrants further discussion.

Response:

We thank the reviewer for the comment. As the responsivities have different baseline values, it might not be straightforward to study the curves in Fig. 2d. To clarify, we have normalized the responsivities shown in Fig. 2d and show the relations in Supplementary Fig. 13. It seems that these three curves have similar trends, implying that the seemingly stronger power dependence of tBLG responsivity should mainly come from its high baseline value (higher responsivity values at each point). In fact, this dependence trend has been observed before [1-5] and ascribed to the sublinear electron heating response under illumination [6]. tBLG has higher absorption, which gives rise to higher electron temperature (can be seen as calibration of the photocarrier density). Therefore, in principle, tBLG should have slightly stronger power dependence due to the sublinear electron heating. This might explain the slightly stronger dependence as shown by red curve in Supplementary Fig. 13.

To clarify, we have revised the main text and added discussion in the Supplementary Note 6 as follows:

Main text, page 6, line 156:

“...We note that the decrease of photoresponsivity with increasing P_{in} is possibly attributed to nonlinear electron heating process [42, 43] and is further discussed in Supplementary Note 6 and Supplementary Fig. 13...”

Supplementary Information, page S11:

“...We have normalized the responsivities shown in Fig. 2d and show the relations in Supplementary Fig. 13. It seems that these three curves have similar trends, implying that the seemingly stronger power dependence of tBLG responsivity should mainly come from its high baseline value (higher responsivity values at each point). In fact, this dependence trend has been observed before and ascribed to the sublinear electron heating response under illumination. tBLG has higher absorption, which gives rise to higher electron temperature (can be seen as calibration of the photocarrier density). Therefore, in principle, tBLG should have slightly stronger power dependence due to the sublinear electron heating. This might explain the slightly stronger dependence as shown by red curve in Supplementary Fig. 13...”

In addition, we have included Fig. R2 in the supplementary Information as Supplementary Fig. 13.

Fig. R2 | Normalized power-dependent responsivities of different graphene devices as indicated by the legends. This figure uses the same data as those in Fig. 2d in the main text.

References:

- [1] M. Freitag et al. *Nat. Photon.* **2013**, 7, 53-59;
- [2] M. W. Graham et al. *Nat. Phys.* **2013**, 9, 103-108;
- [3] P. Ma et al. *ACS Photon.* **2019**, 6, 154-161;
- [4] J. Guo et al. *Light. Sci. Appl.* **2020**, 9, 29;
- [5] S. Schuler et al. *Nat. Commun.* **2021**, 12, 3733;
- [6] K. J. Tielrooij et al. *Nat. Nanotech.* **2015**, 10, 437-443;

Question 5:

How is the bias dependent speed performance of the device?

Response:

We thank the reviewer for the question. To study this, we have fabricated new device and carried out new experiments whose results and discussion are added as **Supplementary Fig. 15** and in **Supplementary note 8**.

The intrinsic photocarrier relaxation time of graphene is in picosecond scale, which sets the theoretical bandwidth to several hundreds of giga Hertz. However, in reality, the speed (or the bandwidth) is limited by the Resistor-Capacitor (RC) value of the device and the measurement circuit. Therefore, in principle as long as the photocurrent signal

is measurable and the RC value is unchanged, the bandwidth should be independent of source-drain bias. This can be evidenced by the similar bandwidths under different biases in Supplementary Fig. 15. The 3dB bandwidth keeps at about 65 GHz under bias voltages of 0.2 V, 0.5 V and -0.5 V. A detailed RC analysis of tBLG device can be found in the Supplementary Note 8.

Fig. R3 | High-frequency response of waveguide-integrated tBLG photodetectors. (a), Optical image of the device, scale bar: 10 μm . (b), Bandwidth data of the device in (a). Green, blue and orange lines denote measurement at bias voltages of 0.2 V, 0.5 V and -0.5 V, respectively. 3-dB cutoff is marked by the red dash line.

Thanks to the reviewer’s question, we have included Fig. R3 in the supplementary Information as Supplementary Fig. 15. The other revision is described as follows.

Main text, page 7

The S_{21} parameter, which representing photoelectrical response, maintains above -3dB from 1 GHz to 65 GHz (measurement limit of our vector network analysis VNA), indicating that the 3 dB bandwidth ($f_{3\text{dB}}$) is beyond 65 GHz (Fig. 3c and Supplementary Fig. 15)

Supplementary Information, page S11:

“...Therefore, in principle as long as the photocurrent signal is measurable and the RC value is unchanged, the bandwidth should be independent of source-drain bias. This can be evidenced by the similar bandwidths under different biases in Supplementary Fig. 15. The 3dB bandwidth keeps at about 65 GHz under bias voltages of 0.2 V, 0.5 V and -0.5 V...”

Question 6:

Clarification is needed regarding the consistently lower bandwidth (~ 36 GHz) observed in eight large-area integrated tBLG photodetectors (Fig. 4d) compared to the device exhibiting a better bandwidth (~ 65 GHz) shown in Fig. 3c. Is the device achieving 65

GHz exceptionally fast compared to others?

Response:

We appreciate the important question from reviewer. As we have discussed in the Response of last question, the bandwidth of a graphene device depends on the RC value, instead of intrinsic photocarrier relaxation time. Therefore, the new device as shown in Fig. R3 shows similar bandwidth with that shown in Fig. 3c as they are fabricated through the same nano-fabricating process, which usually gives similar RC, or more specifically, similar contact resistance.

However, for large-scale integration and fabrication, the urgent task is achieving wafer-scale integration with uniform device performances. This demands the large graphene film has very limited cracks, very few ripples and folds, which remain a substantial challenge. In our work, we not only demonstrate a reliable method to fabricate large-scale tBLG and transfer it onto silicon photonic substrate, but also show the uniform performances including responsivity and bandwidth. We hope these evidences have been able to confirm that tBLG is a potential candidate for detectors in silicon photonics. With the uniform performance as precondition, further in-depth study on improving graphene/metal contacts with CMOS-compatible metals (Ni, Cu, etc.) is needed [1], in order to minimize RC constant in a scalable scale and to increase the bandwidth.

Despite the relatively lower bandwidth of ~ 36 GHz, the detector arrays are still able to show eye diagrams for at 40 Gbit s^{-1} on-off keying (OOK) signal as shown in Fig R4. This confirms the reliability of large-scale integration and fabrication of tBLG.

Fig. R4 | Data reception testing of large-area integrated tBLG photodetectors. Obtained eye diagram for 40 Gbit s^{-1} on-off keying (OOK) data stream of PD 1 (a) and PD 2 (b). Scale bar: 5 ps.

To clarify, we have added discussion in the main text as follows:

Manuscript, page 7:

“...In our work, we not only demonstrate a reliable method to fabricate large-scale

tBLG and transfer it onto silicon photonic substrate, but also show the uniform performances including responsivity and bandwidth. This confirms the reliability of large-scale integration and fabrication of tBLG...”

In summary, we hope that we have addressed all the comments and we thank again the reviewer's constructive suggestions, which have greatly helped us to improve our manuscript.

References:

[1] M. Romagnoli, et al. *Nat. Rev. Mater.* **2018**, 3, 392-414.

Response to the 3rd Reviewer

Qinci has conducted a detailed study on waveguide-integrated twisted bilayer graphene photodetectors, achieving a record-high photoresponsivity of 0.65 A/W for the telecom wavelength of 1550 nm. This breakthrough was made possible by the enhanced optical absorption in twisted bilayer graphene, facilitated by van Hove singularities in the band structure with a 4.1° twist angle. The study showcases significant advancements in photodetector performance, including over 65 GHz bandwidth and 50 Gbit/s data transmission rates, promising for future opto-communications and silicon photonics integration. I find this study to be a particularly intriguing contribution. I recommend this paper for publication, noting its potential impact. However, there are some areas that require further scrutiny and detailed analysis, as highlighted below:

Response:

We appreciate the positive and insightful comments from the reviewer. The reviewer's constructive suggestions help bring significant improvements in our manuscript. We have addressed the reviewer's comments point by point as follows.

Question 1:

The author posits that This configuration has two electrode/tBLG interfaces at source terminal (as shown by the two red arrows in the inset of Fig. 1a). Both interfaces have potential gradients and are used for separating photocarriers in tBLG, therefore double the total photocurrent. This assertion presumes a direct correlation between photocurrent enhancement and graphene's optical absorption capabilities. However, the claim that this configuration doubles the total photocurrent necessitates a more comprehensive discussion. Specifically, it would be beneficial to elucidate the mechanisms through which these interfaces enhance photocarrier separation and photocurrent generation. Additionally, for comparative analysis, could you specify the reference structure or conditions under which this doubling of photocurrent is observed? This clarification would provide a clearer understanding of the efficiency and effectiveness of the proposed design in photodetector applications.

Response:

We thank the reviewer for this insightful suggestion. The GND–S–GND design has been used in literatures [1-3]. The assertion of it allows a doubling of total photocurrent if compared to the simple GND–S case” was firstly noted in Ref [1], but without strict experimental evidence. In theory, the doubling of either light absorption or the separation efficiency could double the total photocurrent. In experiments, it is hard to verify that one interface with gradient is 50% less than the GND-S-GND design because one interface means the whole source-electrode metal covers half of the waveguide,

which will cause additional loss and make the comparison invalid.

If we assume the absorption would keep the same, the doubling-photocurrent concept might be correct if the separation by the two interfaces is independent from each other. This might not true because photocarriers diffuse and the distance of the two interfaces must be taken into account. In the end, it is hard to design a control device with the same resistance, absorption, and interfacial gradient, and further systematic study is necessary. Therefore, we have revised the sentence and delete this assertion. The revision is shown as follows.

Manuscript, page 4, line 74

“...Both interfaces have potential gradients and are used for separating photocarriers in tBLG, therefore enhance the photocurrent...”

References:

[1] A. Pospischil, et al. *Nat. Photon.* **2013**, 7, 892–896.

[2] J. Guo, et al. *Light: Sci & Appl.* **2020**, 9, 29.

[3] F. Xia, et al. *Nat. Nanotech.* **2009**, 4, 839–843.

Question 2:

Could the author provide a more detailed analysis comparing the absorptive properties of the 4.1° twisted bilayer graphene (tBLG), the AB-stacked bilayer graphene, and single-layer graphene (SLG)? The manuscript notes that the absorption curve for the 4.1° tBLG reaches saturation more rapidly than those of the other two materials. This observation suggests underlying differences in optical properties and electron-band interactions among these graphene configurations. It would be insightful to understand the mechanisms driving this distinct behavior of the 4.1° tBLG, particularly in terms of its band structure and interaction with light. Further elucidation on how these differences impact the overall performance and efficiency of photodetectors employing these materials would also be valuable.

Response:

We thank the reviewer’s suggestion in improving the discussion. Unlike the linear band structure of single layer graphene (SLG) and parabolic-like band structure of AB stacked bilayer graphene (BLG), the Dirac cones of the two individual monolayers in tBLG intersect and form saddle points in reciprocal space, resulting in the formation of van Hove singularities (VHSs) in the density of state (DOS)[1,2]. In addition, the position of the intersection (and thus VHSs) is θ -dependent. The energy difference between the VHSs can be described as $\Delta E_{\text{VHS}} = 2\hbar v_{\text{F}} K \sin(\frac{\theta}{2})$. When the incident

P
A
G
E

photon energy matches ΔE_{VHSs} , a pronounced interband transition between VHSs and an enhanced absorption happen [3]. This can be understood by the joint density of states (JDOS):

$$JDOS(\omega) = \frac{1}{4\pi^3} \int \delta[E_c(\mathbf{k}) - E_v(\mathbf{k}) - \hbar\omega] d\mathbf{k} \quad (\text{R1})$$

Where E_c , E_v , \mathbf{k} and \hbar are conduction band energy, valance band energy, reciprocal vector, and photon energy, respectively. The integration indicates that the JDOS is highly relevant to the density of states (DOS). For tBLG, when these two singularities match the photon energy, which means the interband transition matches the delt function in the equation R1, the absorption is enhanced. For example, the photon energy of 0.8 eV corresponds to an incident λ of 1,550 nm, θ is thus estimated to be $\sim 4.1^\circ$.

As suggested by the reviewer, we have included the discussion on enhanced absorption of tBLG in the Supplementary Note 1 and Supplementary Fig. 3 in the Supplementary Information.

Supplementary Fig. 3 | The energy band diagrams (left) and density of states (right) of tBLG(a), SLG(b), and AB-stacked BLG(c), respectively. The tBLG yields a unique optical absorption feature associated to θ .

Supplementary Information, page S3:

Unlike the linear band structure of single layer graphene (SLG) and parabolic-like band structure of AB stacked bilayer graphene (BLG), the Dirac cones of the two individual monolayers in tBLG intersect and form saddle points in reciprocal space, resulting in the formation of van Hove singularities (VHSs) in the density of state (DOS)[1,2]. In addition, the position of the intersection (and thus VHSs) is θ -dependent. The energy difference between the VHSs can be described as $\Delta E_{\text{VHS}} = 2\hbar v_F K \sin(\frac{\theta}{2})$. When the incident photon energy matches ΔE_{VHSs} , a pronounced interband transition between VHSs and an enhanced absorption happen [3]. This can be understood by the joint density of states (JDOS):

$$JDOS(\omega) = \frac{1}{4\pi^3} \int \delta[E_c(\mathbf{k}) - E_v(\mathbf{k}) - \hbar\omega] d\mathbf{k} \quad (\text{S4})$$

Where E_c , E_v , \mathbf{k} and \hbar are conduction band energy, valance band energy, reciprocal

vector, and photon energy, respectively. The integration indicates that the JDOS is highly relevant to the density of states (DOS). For tBLG, when these two singularities match the photon energy, which means the interband transition matches the delta function in the equation R1, the absorption is enhanced. For example, the photon energy of 0.8 eV corresponds to an incident λ of 1,550 nm, θ is thus estimated to be $\sim 4.1^\circ$.

References:

- [1] W. Landgraf, et al. *Phys. Rev. B* **2013**, 87, 075433.
- [2] J. Yin, et al. *Nat. Commun.* **2016**, 7, 10699.
- [3] K. Yu, et al. *Phys. Rev. B* **2019**, 99, 241405(R).

Question 3:

Supplementary Figure 3 is missing essential elements, specifically scale bar/color bar. These components are crucial for interpreting the figure's data accurately. Could the author please address this oversight by adding the missing scale and color bars to Supplementary Figure 3? Including these elements would significantly enhance the figure's clarity and enable readers to better understand the presented data.

Response:

We thank the reviewer's careful examination and comments. We have added scale and color bars in the **Supplementary Figure 3 (currently the Supplementary Figure 4)** to describe the data more clearly, and the revised figure was as below.

Supplementary Fig. 4 | Simulated electric-field profile of the TE waveguide mode. The field distribution along the tBLG sheet is shown as the red line. A strong field localization occurs around the metal/tBLG interface, which is greatly beneficial for the

highly efficient photon absorption.

Question 4:

In the supplementary section, it is stated that 'The contact resistance is estimated as $\sim 500 \Omega \cdot \mu\text{m}$ (Supplementary Fig. 6), which suggests a unit of resistivity rather than resistance. Typically, contact resistance is measured in ohms (Ω), while ohm-micrometers ($\Omega \cdot \mu\text{m}$) are a unit of resistivity. This discrepancy might lead to confusion about the nature of the measurement being reported. Could the author please clarify this point and review the units used to ensure they accurately represent the quantities being measured? Correcting this could enhance the precision and reliability of the data presented.

Response:

We thank the reviewer for pointing out the mistake we made. We have changed the sentence “The contact **resistance** is estimated as $\sim 500 \Omega \cdot \mu\text{m}$ ” into “The contact **resistivity** is estimated as $\sim 500 \Omega \cdot \mu\text{m}$.”.

In the experiment, we measured the sum of two contact resistances is about 125Ω , so one contact has resistance as 62.5Ω . As the wider a device is, the smaller the contact resistance becomes. The width of the contact is $8 \mu\text{m}$, so the normalized resistivity is $62.5 \Omega \cdot 8 \mu\text{m} = 500 \Omega \cdot \mu\text{m}$.

Question 5:

The statement regarding the photodetectors having a relatively smaller footprint due to enhanced absorption, attributed to the unique van Hove singularities (vHs) of twisted bilayer graphene (tBLG), alongside the intensified field intensity at the electrode/tBLG interfaces as illustrated in Figure 1d, raises a question about the mechanism of field enhancement. Is the field enhancement primarily attributed to the plasmonic mode induced by the interaction between the metal and the waveguide or specifically by the tBLG itself? Clarifying the primary source of field enhancement would provide deeper insight into the operational dynamics of the photodetectors and the role of tBLG in enhancing their efficiency.

Response:

We are grateful for the reviewer’s instructive comments. As revealed by our previous work, the absorption enhancement of tBLG is compatible to absorption of field enhancement from plasmonic effect [1], which means these two enhancement factors can be multiplied to give rise larger enhancement. In principle these two processes are independent: The field enhancement can be seen as “concentration of optical field (or

P
A
G
E

optical beam)” as the light intensity equals the square of the optical electric field”. It happens at the optical illuminating process. On the other hand, the absorption enhancement of tBLG happens at the electron excitation process and is due to the enhanced joint density of states (or van Hove singularities). An evidence of the enhancement-multiplication process can be seen in Ref [1].

Thanks to the reviewer’s comment, we have added the discussion in the Supplementary note 2, as follows:

Supplementary Information, page S5:

The absorption enhancement of tBLG is compatible to absorption of field enhancement from plasmonic effect, which means these two enhancement factors can be multiplied to give rise larger enhancement. In principle these two processes are independent: The field enhancement can be seen as “concentration of optical field (or optical beam)” as the light intensity equals the square of the optical electric field”. It happens at the optical illuminating process. On the other hand, the absorption enhancement of tBLG happens at the electron excitation process and is due to the enhanced joint density of states (or van Hove singularities).

References:

[1] J. Yin, et al. *Nat. Commun.* **2016**, 7, 10699.

REVIEWERS' COMMENTS

Reviewer #1 (Remarks to the Author):

In my opinion the authors have addressed all the questions and comments of the reviewers. Even though the results may address new questions on performances, the basic demonstration of high responsivity with 4.1° twisted bilayer graphene is still the main value of the work. I'm sure that by improving the technology the performances can be better and I understand that the authors were limited by the setup. I'm satisfied of the revised version and I suggest to accept this manuscript for publication in Nature Communications.

Reviewer #2 (Remarks to the Author):

I am satisfied with the author's revisions and responses, and have no further questions.

Reviewer #3 (Remarks to the Author):

After reviewing the revised version and the author's response, my questions and comments have been adequately addressed and explained. The revisions have improved the clarity and completeness of the manuscript. I recommend the publication of this paper in Nature Communications.